# Systemic Risks of Climate Events and Households' Participation in Mariculture Mutual Insurance: A Case Study of Shrimp Producers in Zhejiang Province

**Hongyun Han [1],\* and Ye Jiang [2]**

[1] Center for Agricultural and Rural Development, School of Public Affairs, Zhejiang University, Hangzhou 310058, China

[2] School of Management, Zhejiang University, Hangzhou 310058, China; jy11320049@zju.edu.cn

\* Correspondence: hongyunhan@zju.edu.cn

**Abstract:** Mariculture is playing an important role in food safety, acting as strong complement to marine fishery. As a typical capital intensive and high-risk sector, mariculture mutual insurance is important for ensuring the stability and sustainability of mariculture due to the inertia of private insurance, it is necessary to examine factors for low household participation in marine fishery mutual insurance to promote the healthy development of marine insurance. Based on the field surveyed data of mariculture shrimp producers in Zhejiang Province, this study aims to examine the determinants underlying households' participation in mariculture mutual insurance. Based on logistic model, we find out that climate risks, environmental risks and technical risks have seriously hindered the development of food security and fisheries in Zhejiang Province. In addition, farmers' insurance involvement mainly depends on the individual characteristics of the farmers: whether used to go out to work, perception of burden level of premium and insurance awareness; family characteristics of fish farmers: total household income, and unpaid loan; and production characteristics: professional level, mariculture area and whether sea waters registration. Meanwhile, external factors, including organizations available for insurance participation, impact of national insurance subsidies, policy support and disasters on the aquaculture area. Corresponding risk management measures are urgently needed for the sustainable development of mariculture.

**Keywords:** systemic risks; climate events; mariculture mutual insurance; Zhejiang

## 1. Introduction

It is believed that the world is experiencing major environmental changes as a consequence of human activities, the most important of which is climate event. Climate event could have a range of related outcomes, including shifting patterns of agricultural production, storm and flood damage, desertification, water shortages [1], and loss of ecosystem resilience. More than three-quarters of recent economic losses caused by natural hazards can be attributed to windstorms, floods, droughts, and other climate-related hazards, which appear to be increasing at a greater rate than geophysical disasters [2]. Coastal communities are more vulnerable to the impacts of climate event [3]. Some of the highlighted impacts include ocean acidification that can lead to reduced sperm motility [4], ocean warming as a result of sea temperature increases [5], and sea-level rise and habitat degradation [6,7].

Agricultural insurance faces 10 times higher risk than other types of normal insurance [8]. Various forms of insurance mechanisms have been developed in developing countries [9]; however, these mechanisms can incur high opportunity costs in the form of foregone development [10]. Therefore,

the expectation of premium compensation and production security level arising from agricultural insurance will affect the farmers' participation behavior [11]. Given the fact that the cost of other ways of diversifying risk transfer is mostly lower than purchasing insurance [12–16], which will definitely inhibit the willingness of fishermen to insure [17]. Private markets for crop insurance are limited by the substantial systemic risk [18,19], which result in low catastrophe insurance penetration of the private insurance sector in developing countries [2]. Mariculture is facing higher risk and more severe natural disasters than other industries [20], such as drought and extreme temperatures during critical periods, disease and infection intensified by adverse weather [21], and ocean disasters [22]. Systemic risk is stemming primarily from the unfavorable weather events of droughts or extreme temperatures [8,23], hurricanes, floods, earthquakes, pervasive freezes, and major snowstorms [24]. The systemic risk is the primary cause of insurance market failure in terms of low participation rate [8], because private insurance markets are only good at handling independent risks-risks that are not correlated across insureds. But insuring catastrophic risk is not so easy [18].

The failures of crop insurance markets in the form of high loss ratios, low participation rates, and the aversion of private insurance companies to bearing exposures have been documented extensively [23]. Creating the right incentives towards increasing farmers' participation in crop insurance has been one of the major goals of the U.S. farm policy in recent years. Although insured acres increased in the 1990s, only one-third of farm producers participated in the crop insurance program [25]. Due to its character of a quasi-public goods with insufficient competitiveness and exclusivity [26], mutual insurance has been developed as an important complementation of the fishery compensation system in Japan and South Korea for a long time [27]. It is important to the sustainable mariculture development to promote the participation of fishermen in policy-based fishery mutual insurance.

At present, research on insurance is mainly concentrated in the field of planting, with less attention on fishery insurance, especially mariculture [28–30]. Existing research on systemic risks concentrates primarily on identifying the nature and magnitude of systemic risks [8,31] and on investigating ways in which the risks can be managed by utilizing private reinsurance and capital markets [8,23,31,32], while weather-based index insurance adaptation measures are being scaled up with a view to assist farmers to adapt to the changing climate-induced disasters [33]. "The 12th Five-Year Plan for Policy-based Fisheries Mutual Insurance in Zhejiang Province" showed that the main fishery mutual insurance in Zhejiang Province is insurance for fishing, including fishing boat mutual insurance and employer liability mutual insurance. The aquaculture-related mutual insurance has only been developed and expanded since 2012, including marine aquaculture mutual insurance, fishermen's accidental injury mutual insurance, and fishery infrastructure mutual insurance. Therefore, little attention has been given to the mariculture fisheries sector [6] and no empirical investigation of mariculture insurance participation has been conducted to date [23].

What are the exact factors determining fish farmers' participation in mariculture insurance, and to what extents? All these issues are important for effective policy design to ensure stable and sustainable mariculture sector. This study is organized as follows: firstly, it is a brief introduction of the development of fishery insurance in Zhejiang Province, which provides a background for a better understanding of fishery insurance in China; then, a model is constructed based on a literature review, which is a theoretical explanation for the selection of variables for our empirical analysis. Thirdly, the regression results are deeply analyzed, it is followed by a brief conclusion, policy implications are given in this section.

## 2. The Development of Fishery Insurance in Zhejiang Province

### 2.1. The Development of Fishery Production and Disasters in Zhejiang Province

Aquaculture is one of the fastest growing food animal sector, accounting for nearly 40.1% of fin- and shellfish consumed worldwide [34]. More than half of seafood is coming from aquaculture [35–37],

and aquaculture production is expected to be more than doubled to 140 million tones by the year 2050 [38]. Mariculture yield surpassed sea-fishing for first time in 2004 all over the world. As the biggest producer of aquatic products, China's fish farming output has exceeded the amount of fishing since 1998 as a result of a series of policies [39]. The culture-based fishery policy was proposed in China's 11th five-year plan in 1985 facilitated the development of mariculture. In 1997, the new fisheries policy of "Developing Aquaculture and Protecting and Rational Utilizing the Sea Fishery Resources" was released to further upgrade fisheries industry structure. Data from China fishery statistical year shows that up to 2015, mariculture production accounted for 28.00% of the total aquatic products and 55.02% of marine products in China. The farming area and output values of mariculture were up to 27.38% and 35.50% of the national aquaculture, respectively. Mariculture is of great significance for ensuring China's food security.

With the expansion of the scale of mariculture as well as the increase of the level of mariculture intensification, disease disasters in marine aquaculture have become one of the main factors restricting the development of mariculture. The loss of fishery yield and value in Zhejiang Province caused by disease increased recently. In the meantime, the mariculture area affected by drought, typhoons and floods had also raised (See Figures 1 and 2). In addition, environmental pollution was also an important factor influencing the development of mariculture. With the development of industrialization and urbanization, the endogenous pollution, brought about by external pollution and the mariculture itself, restricts the development of mariculture. But mariculture losses caused by pollution is still smaller than climate disasters and disease (See Figure 1).

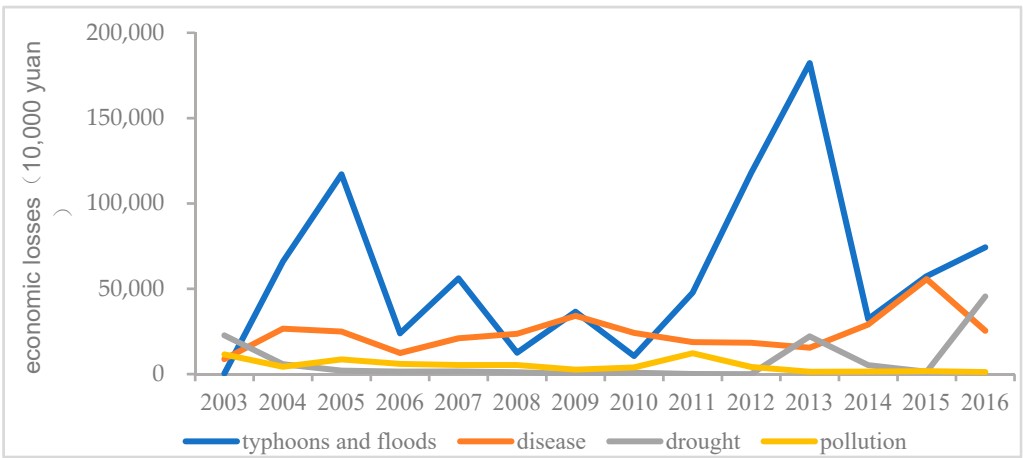

**Figure 1.** Fishery economic loss caused by various disasters in Zhejiang Province from 2003 to 2016. (Source: China fishery statistical yearbook, 2004–2017, Fisheries and Fisheries Administration of Ministry of Agriculture and Rural Affairs of the People's Republic of China.).

Disasters mainly include typhoons, floods, droughts, pollution, and diseases. The following figures are presentation of different disasters happened in China. Based on data from China fishery statistical yearbook, as the main contributor of economics loss, typhoons is fluctuating over time, followed by diseases, pollution, and droughts. If an affected area is termed as an aquaculture area whose yield reduction is more than 10% resulted from different kinds of disasters, the main disaster is typhoons, which is followed by diseases, droughts, and pollution (See Figure 2).

The regional distribution of mariculture has transformed from traditional production areas to all coastal provinces, including Liaoning, Hebei, Shandong, Jiangsu, Zhejiang, Fujian, Guangdong, Guangxi, Hainan, and Tianjin. As a traditional production base, Shanghai quitted mariculture in 2009 as a result of industrial structural adjustments. Figure 3 shows that Shandong is the biggest contributor of maricultural value in China.

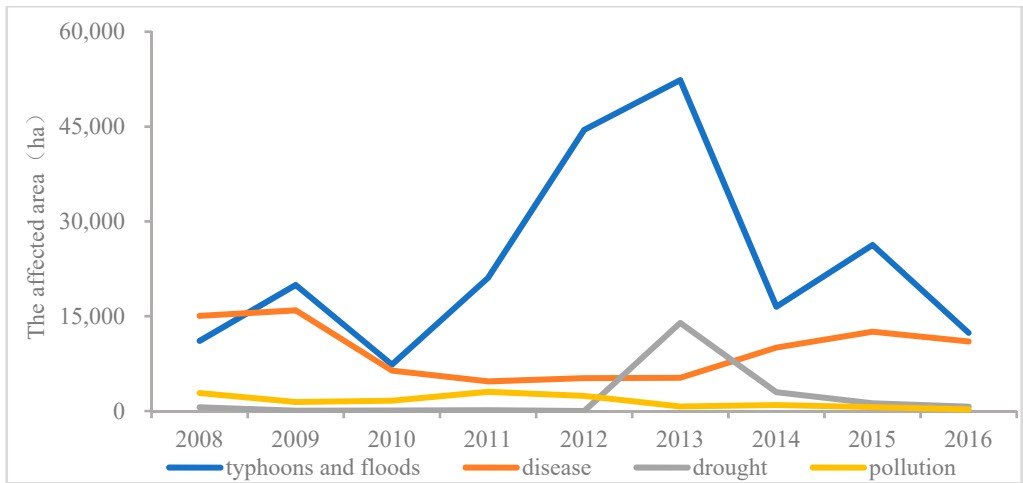

**Figure 2.** The area affected by various disasters in Zhejiang Province from 2008 to 2016. (Source: China fishery statistical yearbook, 2009–2017, Fisheries and Fisheries Administration of Ministry of Agriculture and Rural Affairs of the People's Republic of China.).

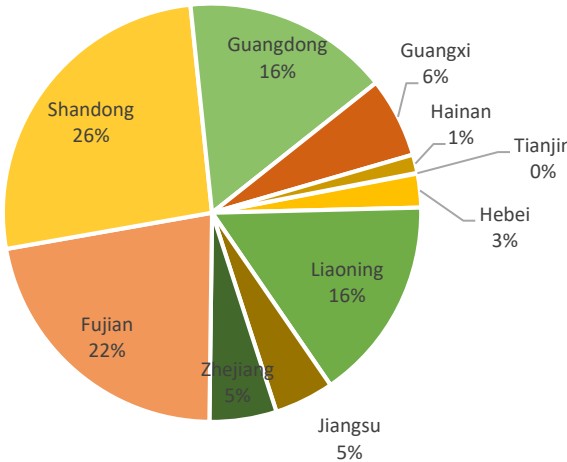

**Figure 3.** The regional distribution of mariculture production in China in 2016. (Source: China fishery statistical yearbook, 2017, Fisheries and Fisheries Administration of Ministry of Agriculture and Rural Affairs of the People's Republic of China.).

From this perspective, Zhejiang is only ranked sixth place (see Figure 3) and its mariculture production contributed 2.3% of agricultural output value (see Figure 4). As a main coastal mariculture province of China, Zhejiang Province takes obvious advantages of mariculture development with long and twisting coastlines, lots of islands, broad shallow sea, bay and tidal flats. It tops the whole country with the 1840.07 km of mainland coastline and 4301.21 km of island coastline, which demonstrates a high percentage of usable waters in China. Data from China fishery statistical yearbook shows that 88,816 ha of waters were used in mariculture in 2016 in Zhejiang, achieving an mariculture output of 1,017,702 tons which accounted for 16.83% of total fishery output, as well as a production value of 14.43 billion Yuan which took 20.92% of the total production value.

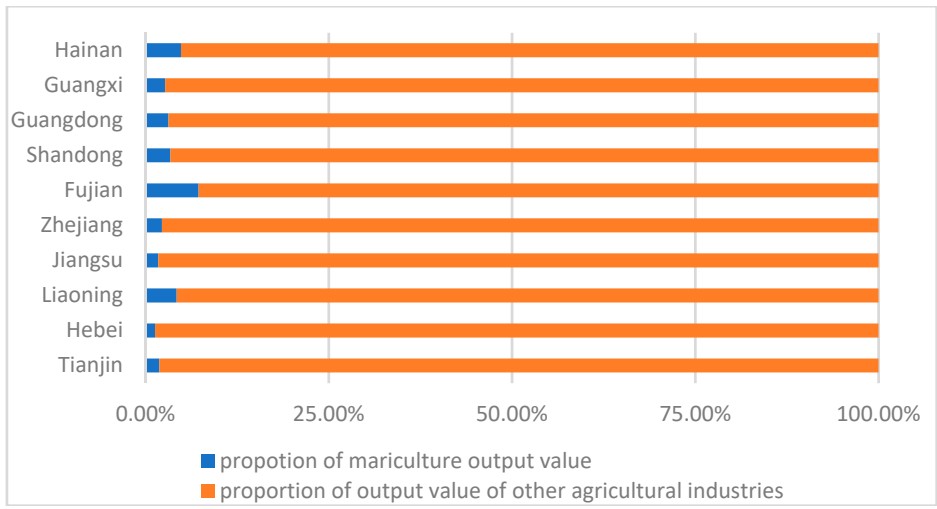

**Figure 4.** Proportion of mariculture output value in different mariculture areas in China in 2016. (Source: China fishery statistical yearbook, 2017, Fisheries and Fisheries Administration of Ministry of Agriculture and Rural Affairs of the People's Republic of China.).

Zhejiang Province has as many as 37% of the islands of the whole country, leading to higher influence on the mariculture by climate disaster. Data from China Marine Disaster Bulletin shows that Zhejiang Province takes the first place in the ratio of being affected by climate disasters, with 27,440 ha of mariculture area hit at a percentage of 31.95%. Data from China Fishery Statistics Yearbook shows that the loss of aquatic products in Zhejiang Province caused by typhoons and floods alone is as high as 12,371 tons, worth 74.32 million Yuan.

### 2.2. The Development of Fishery Insurance in Zhejiang Province

In 1994, China Fishery Insurance Association set up an office in Zhejiang Province, and Zhejiang Fishery Mutual Insurance Association was established in 2004. In 2005, the pilot project of policy fishery insurance was extended at six cities of Wenzhou and Taizhou of Zhejiang Province, with 1.23 million Yuan from provincial financial subsidy fund and 758,000 Yuan from local governments. In 2005, Zhejiang Province took the lead in launching the fishery mutual insurance premium subsidy, and pioneered the exploration of policy fishery insurance system. According to the risk management policy of "prevention–compensation combination" in China, a series of measures such as pre-insurance check, safety publicity, disaster warning, and implementation of disaster prevention technologies have been proposed and implemented [40,41].

In 2008, the Zhejiang Provincial Government officially incorporated policy fishery insurance into agricultural insurance, and issued the "Interim Measures for the Administration of Special Funds for Policy Fishery Insurance Subsidies in Zhejiang Province". In 2013, Central Committee's Document No. 1 clearly stated that "the pilot of fishery insurance premium subsidies" focused on the principle of "mutual assistance" and served fisheries with a cover of fishing boat property and personal accidents. Twenty-three city offices and three service centers have been established in Zhejiang Province.

At present, the insured fishery products mainly include shallow sea shellfish, seawater pond crabs, sea fish, shallow seaweeds, seawater shrimps, turtles, and other famous aquatic products. Among them, insured products for shrimp farming include seawater ponds, high-level ponds, and high-density greenhouses. The "Twelfth Five-Year Plan for Policy Fishery Mutual Aid Insurance in Zhejiang Province" in 2011 clearly proposed the goal of promoting the scale of various mutual insurances. It was estimated that in 2015, the coverage of fishery insurance would reach 70% of the aquaculture area, and the mutual insurance premium would be raised from 90,000 Yuan to 80 million Yuan, with a total underwriting amount of 5 billion yuan. The participants of fishermen's accidental injury mutual insurance reached 25,000 persons in 2015, accounting for 60% of the total number.

The average amount of accidental injury mutual insurance reached 250,000 yuan/person, with the total underwriting risk insurance at 6.5 billion yuan and insurance premium at 15 million Yuan per year in 2015.

In 2013, the Provincial Party Committee and the Provincial Government's "Opinions on Promoting the Continued and Rapid Growth of fishermen's Income" proposed "expanding the pilot of fishery mutual insurance to aquaculture", which facilitated the development of mariculture. It was also estimated that in 2015 the insurance participation rate of government investment public infrastructure such as fishing port breakwaters, wave dams and fishing piers reached 80%, and the rate of collective property such as fishing boat repair enterprises reached 50%, which basically covered the repair of fishery infrastructure damage caused by climate disasters, with the total underwriting risk compensation to be 6 billion Yuan and the total annual mutual insurance premium reaching 40 million Yuan.

The types of liability mutual insurance business mainly include: mutual protection of fishing vessels (full damage liability, full damage additional collision third party ship liability, comprehensive liability, comprehensive additional third-party personal injury liability, comprehensive additional fishing gear responsibility, comprehensive additional damage, and full liability), employer liability mutual insurance, employer liability plus accidental injury medical mutual insurance. The types of mutual insurance carried out in the pilots are: deep-water cages and additional breeding responsibility mutual insurance, and leisure fishery passenger accidents.

Mutual insurances include mutual protection of injury, fishery infrastructure, fishermen's microfinance borrowers, and accidental injuries of fishermen's microfinance borrowers. Aquaculture mutual insurance is an important part of building a fishery risk protection system.

China Fisheries Mutual Insurance Association adopted the "cooperative organization + administrative assistance" mode of operation in actual operation. Since 2013, the Zhejiang Provincial Fisheries Mutual Insurance Association has followed the guiding principles of "guaranteeing disasters, guaranteeing costs, and ensuring the recovery of production capacity" in accordance with the principles of "government guidance, market operation, voluntary, and mutual assistance among members". In 2016, Zhejiang undertook a total of 628 orders, providing risk protection of 466 million yuan, 15 times more than that of 29 million yuan at the beginning of the pilot in 2013.

Fishery insurance is proved to be the most effective tool for the systemic risk transfer of mariculture [42–44] in ensuring fishermen's income and the coherence of fishery production [40], enhancing the financing capacity of fishermen to expand reproduction [45], improving the international competitiveness of agricultural products as the "Green Box" support policy [46], driving new technologies and developing new production methods [46], promoting the industrialization and modernization of fisheries [41,47], and strengthening farmers' confidence in adopting new technologies, introducing new varieties and new production methods [46]

The private agricultural insurance market does not work efficiently due to the quasi-public product attributes and positive externalities of agricultural insurance. As a form of risk sharing between members, mutual insurance is expected to cater to smaller groups of homogenous members [48]. The place of fishery mutual insurance and the establishment, operation, supervision, and preferential policies of institutions are subject to further reform and improvement [49]. Data from China Fishery Mutual Insurance Association showed that the annual comprehensive compensation rate of the China Fisheries Mutual Insurance Association is above 40% on average, with a high comprehensive compensation rate and low efficiency. In the event of a major disaster, it may even be unable to make ends meet [50]. In next section, we examine factors that influence fishermen's decision to participate in the fishery mutual insurance under various economic and policy scenarios, and discuss different ways of creating incentives to increase and diversify the insured pool of participants in the insurance markets.

## 3. The Model and Data

### 3.1. The Model

Kunreuther [51] and Slovic et al. [52] found that the decisions to buy insurance are consisted of three phases: awareness of disasters and potential losses, take insurance as a tool to deal with disasters, and obtain insurance information and process information [53]. The individual characteristics and family characteristics of farmers are important influencing factors of insurance participation behavior selection. Ye [54] stated that risk level, income, premiums, insurance awareness, and subsidies would all affect demand. Jia and Chen [55] investigated the factors influencing demand for aquaculture insurance, concluding that fish farmers' age, income, previous losses, compensation and insurance knowledge were positively correlated with demand, but premiums exhibited an opposite correlation. Wu [56] used the data from 130 fish farmers in Hubei Province and identified the key factors influencing demand for the freshwater aquaculture insurance. Those factors are fish farmers' age, income, fishery species, and financial loss over the years. Some scholars found that the age and education level of farmers had no significant influence on their willingness to participate in the insurance program [11], while some scholars also found that the willingness of farmers to participate in the insurance program would increase with the decrease of age and the increase of education level [57]. Holthausen and Baurfound [58] that the family income structure had a significant impact on the participation of farmers in the insurance. When the agricultural income was not the main economic source of the family, the farmers were not willing to buy agricultural insurance. Other risk transfer modes of dispersion will have an inhibitory effect on fishermen's insurance intention [17]. Measures such as differentiated planting, conservative technology, part-time business, and reciprocal credit can help farmers to prevent and share risks at a lower cost than participating in insurance [13,15] Agricultural insurance has the nature of quasi-public products, and pure commercial agricultural insurance is unable to operate, and has to be corrected with the assistance of government subsidies and maintain the normal operation of commercial agricultural insurance [29]. Government subsidies on agricultural insurance premiums have increased the enthusiasm of farmers to participate in insurance [30]. The motivation for farmers to participate in mutual insurance is the expected benefits from premium subsidies by avoiding risks [59]. Through a general survey of the researches on the participation of agricultural insurance, we can find that the participation of farmers is mainly affected by the individual characteristics of farmers, characteristics of family, production characteristics and other influencing factors include farmers' awareness of insurance, insurance subsidies, policy support, etc.

As we want to study the participation behavior of mariculture shrimp fishermen, which is a probability problem. Therefore, A binary discrete selection model is mainly used in this paper to analyze the influencing factors of marine fishermen's participation behavior. The basic form of the Logit model is as follows:

$$p = FZ = 1/\ 1 + e^{-z} \tag{1}$$

In the Equation (1), Z is a linear combination of the variables $X_1, X_2, X_3 \ldots X_n$ that is:

$$Z = b_0 + b_1 x_1 + b_2 x_2 + \ldots + b_n x_n \tag{2}$$

Transform Equations (1) and (2) to obtain the Logit model form expressed in odds:

$$Ln\left(\frac{p}{1-p}\right) = b_0 + b_1 x_1 + b_2 x_2 + \ldots + b_n x_n + e \tag{3}$$

In Equation (3), p is the probability that the fisherman will participate in fishery insurance. $X_i$(I = 1,2, ... ,n) is the explanatory variable; $b_0$ is a constant term, $b_i$ is the regression coefficient of the ith influencing factor; e is a random variable, including possibly unobservable characteristics influencing the final decision. The values of $b_0$ and $b_i$ can be estimated with the maximum likelihood estimation method.

$X_i$(I = 1,2, . . . ,n) is the explanatory variable, they are Age, Education, Experience, Gender, Labors used go out to work, Professional level, Mariculture area, Registration of sea waters, Loss of yield value, Participation in cooperative organization, Awareness of insurance, Perception of burden level of premium, Household total income, and Unpaid productive loan.

These variables are surveyed and measured as follows:

Age is denoted as the age of a mariculture fisherman.

Education is the school years of a mariculture fisherman.

Experience is the years of shrimp farming, which represents technological risk. The longer the fishermen engaged in marine aquaculture, the less technical risks they face.

Gender is the gender of mariculture fishermen.

Labors used go out to work means whether mariculture fishermen used to go out to work.

Professional level is the proportion of shrimp mariculture income to total income.

Mariculture area is the water area used to shrimp farming of a household.

Registration of sea waters means whether the sea area is registered for authority.

Loss of yield value is the loss of shrimp mariculture value per unit area affected by different disasters in recent three years, mainly including typhoon, flood, disease, drought, and pollution. This variable is used to measure mariculture risk, including climate risk and environmental risk. Climate risk is represented by the mariculture loss of yield value influenced by typhoon, flood and drought. Environmental risk is represented by the mariculture loss of yield value influenced by disease and pollution.

Participating cooperative organization means whether mariculture fishermen join fishery cooperative organizations, including cooperatives, cooperation with the company, alliance between large farmer households.

Awareness of insurance means the degree of understanding on fisheries insurance.

Perception of burden level of premium means the feelings of mariculture fishermen about fishery insurance premium.

Household total income means income from sources of mariculture, agriculture, operating income, wage income, asset income, and transfer income.

Unpaid productive loan is measured by the loan need to repaid over the next period of time of mariculture shrimp fishermen household.

All the data are collected by field survey, and then these data will be calculated as mentioned above, logit model will be used to do the regression, and then analyzing the data with a regression.

### 3.2. The Data

In this study, seawater shrimp farmers in Zhejiang Province were selected as research objects, with questionnaires designed and surveyed. "2014 Zhejiang Fishery Statistical Yearbook" showed that the mariculture shrimp production in Ningbo, Taizhou, Zhoushan, and Wenzhou, accounted for 48.24%, 32.19%, 9.8% and 8.3% of total output. The survey on mariculture shrimp was mainly located in four areas: Ningbo, Taizhou, Zhoushan, and Wenzhou. According to the proportion of production in each region in 2014, random sample surveys were conducted among farmers in the four regions. In order to ensure the quality and effectiveness of the questionnaire, all the questionnaires were filled by the members of the research team and the fishermen with one-on-one inquiry.

The distribution of the samples is given in Figure 5 and Table 1, a total of 220 questionnaires in Ningbo, 150 in Taizhou, 80 in Zhoushan, and 60 in Wenzhou. The total 457 effective questionnaires were collected, 199 in Ningbo, 137 in Taizhou, 69 in Zhoushan, and 52 in Wenzhou. The effective rate was about 89.6%, which means the proposition of questionnaires with complete information in total questionnaires.

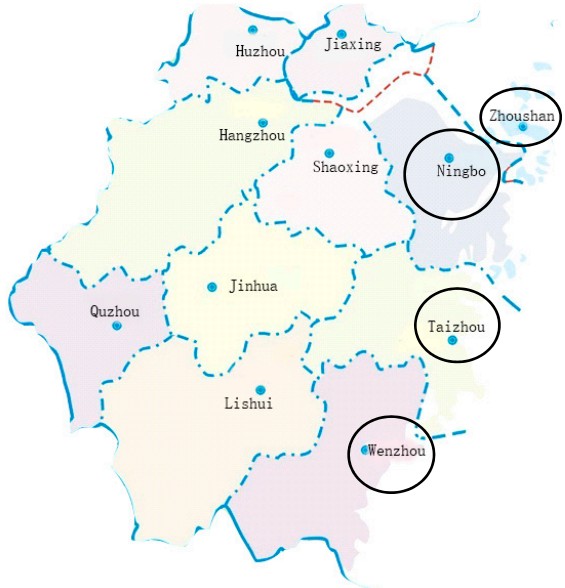

**Figure 5.** Distribution of surveyed samples in Zhejiang Province.

**Table 1.** The investigation and recovery of the data.

| Prefecture-Level City | County | Town | Investigation | Recovery |
|---|---|---|---|---|
| | | Hepu | 78 | 68 |
| Ningbo | Xiangshan | Dingtang | 69 | 63 |
| | | Gaotang | 73 | 68 |
| Taizhou | Sanmen | Shepan | 97 | 88 |
| | | Zhuao | 53 | 49 |
| Zhoushan | Putuo | Taohua | 45 | 40 |
| | | Liuheng | 35 | 29 |
| Wenzhou | Yueqing | Yandang | 60 | 52 |

The dependent variable in this paper is whether the fishermen "takes participation in fishery insurance". In terms of the determination of independent variables, the age of the household, the level of education, the experience of farming, the area of cultivation, the degree of specialization, the loss of unit area of disasters in the past three years, the total income and the unproductive loans are expressed by actual values. The gender of the fishermen, whether to go out for work, registration of sea waters is dummy variables. The awareness of insurance and the perception of insurance burden are measured by a scale.

Table 2 shows there were about 45% of mariculture shrimp farmers participating in fisheries insurance. The systemic risk faced by fishermen is the premise and determinant of their participation in fishery insurance. The fishermen surveyed were affected by the disasters in the past three years, including typhoons, floods, droughts, pollution and diseases. The average loss in Zhejiang Province was as high as 1272 Yuan/Mu. There is even no harvest at all when facing severe disasters. Among different kinds of disasters, the loss of mariculture yield value per unit area influenced by typhoon and flood is highest in Zhejiang. The loss of yield value per unit area influenced by disease, Drought and Pollution also have a great impact on mariculture in Zhejiang.

**Table 2.** Descriptive statistics of variables in regression.

| Abbreviations | Variables | Meaning and Unit | Mean | St.dev | Min | Max |
|---|---|---|---|---|---|---|
| FIP | Fisheries insurance participation | yes = 1, no = 0 | 0.45 | 0.50 | 0 | 1 |
| Age | Age | year | 50 | 7.85 | 25 | 69 |
| Edu | Education | year | 7.49 | 3.23 | 0 | 16 |
| Exp | Experience | year | 10.90 | 5.46 | 0 | 28 |
| Gender | Gender | man = 1, woman = 0 | 0.84 | 0.36 | 0 | 1 |
| GOW | Whether used to go out to work | yes = 1, no = 0 | 0.30 | 0.46 | 0 | 1 |
| PL | Professional level | Income from mariculture shrimp/total household income | 0.64 | 0.32 | 0.14 | 1 |
| Area | Mariculture area | mu | 27.36 | 13.81 | 7 | 85 |
| SWR | Whether sea waters registration | yes = 1, no = 0 | 0.27 | 0.45 | 0 | 1 |
| LYV | Loss of yield value per unit area | Loss of yield value per unit area influenced by different kinds of disasters in recent three years (yuan) | 1826 | 1272 | 215 | 5000 |
| LTF | Loss of yield value per unit area | Loss of yield value per unit area influenced by typhoon and flood (yuan) | 969 | 875 | 0 | 4618 |
| LDro | Loss of yield value per unit area | Loss of yield value per unit area influenced by drought (yuan) | 235 | 514 | 0 | 3968 |
| LDis | Loss of yield value per unit area | Loss of yield value per unit area influenced by disease (yuan) | 358 | 512 | 0 | 3333 |
| LP | Loss of yield value per unit area | Loss of yield value per unit area influenced by pollution (yuan) | 266 | 477 | 0 | 2565 |
| PCO | Whether participate in cooperative organization | yes = 1, no = 0 | 0.38 | 0.49 | 0 | 1 |
| AI | Awareness of insurance | very little = 1, relatively little = 2, general = 3, relatively more = 4, very well = 5 | 3.26 | 0.92 | 1 | 5 |
| PLP | Perception of burden level of premium | very small = 1, relatively small = 2, general = 3, relatively large = 4, very large = 5 | 3.94 | 1.14 | 1 | 5 |
| HTI | Household total income | 10,000 yuan | 49.04 | 24.40 | 15 | 125 |
| UPL | Unpaid productive loan | 10,000 yuan | 6.68 | 9.02 | 0 | 64 |

Among the 457 households surveyed, there were 387 men and 70 women, accounting for 85% and 15% of the total samples. The number of male samples was larger than that of female. Among the samples, 137 households had the experience of going out for work, accounting for 30% of the total subjects. The average age was 50 years old, with the minimum age 25, and the maximum 69. And the median age was 51 years old. The average education level for fishermen was 7.5 years, and the most of education level is middle school (9 years), accounting for 77% of the total subjects. The average household income of fishermen was 490,000 Yuan. The income gap among the samples was large, with lowest income at 150,000 Yuan, and the highest reaching 1.25 million Yuan. The average specialization of household production, as the proportion of production income of sea shrimp culturing to the total, was 0.63. There were households with more polyculture and part-time income, with the minimum specialization of 0.14.

Total aquaculture area is an important indicator of shrimp production. Compared with traditional crop production, mariculture had a larger production area with an average area of 27 Mu. The smallest aquaculture area was 7 Mu, while largest was as high as 85 Mu, which indicated a significant trend of farming scale. Whether the mariculture sea areas are registered or not may affect the decision of fishermen's participation. Among them, 332 subjects had not registered, accounting for 72.65% of the total sample size.

Registration can make clear the scope and stability of the sea waters. There are 27% investigated marine farmers obtain confirmation registration. There are 38% investigated marine farmers participate in cooperative organization. The mean value of awareness of insurance of surveyed marine farmers are about 3, which shows mariculture shrimp fishermen have a general awareness of insurance in Zhejiang.

The mean value of perception of burden level of premium of surveyed marine farmers are around 4, which shows mariculture shrimp fishermen feel relatively large burden of insurance premium.

　　Household total income of surveyed marine farmers various from 150 thousand yuan to 1250 thousand yuan. The average income of mariculture shrimp fishermen family is 490 thousand yuan, which has not deducted farming production cost.

　　There are a lot of mariculture fisher family have unpaid load, and the highest household unpaid loan is 640,000 yuan. The average unpaid loans were about 66,800 Yuan, among which 366 households had unpaid loans of less than 100,000 Yuan (183 have no loans), accounting for 80.1% of the total samples. And there were 60 households with unpaid loans of between 100,000 and 200,000 Yuan, accounting for 13.1% of the total sample, and 6.8% of all had unpaid loans of more than 200,000 Yuan, the highest unpaid loan was 640,000 Yuan (see Table 3).

**Table 3.** Marine farmers' household unpaid loan in 2015.

| Loan (10,000 yuan) | (0,10) | (10,20) | (20,30) | (30,40) | (40,50) | (50,60) | (60,70) | Total |
|---|---|---|---|---|---|---|---|---|
| **Proportion (%)** | 80.09 | 13.13 | 3.28 | 2.41 | 0.44 | 0.44 | 0.22 | 100 |
| **Cumulative proportion (%)** | 80.09 | 93.22 | 96.5 | 98.91 | 99.34 | 99.78 | 100 | |
| total | 366 | 60 | 15 | 11 | 2 | 2 | 1 | 457 |

## 4. Regression Results

### 4.1. Empirical Result

　　Here we use Stata 14 to do the logit regressions. The logit model estimation and test results of marine fishermen's insurance participate on behavior are shown in Table 4.

**Table 4.** Logit Regression results of mariculture insurance participate behavior.

| Variables | Model 1 | | Model 2 | | Model 3 | | Model 4 | |
|---|---|---|---|---|---|---|---|---|
| | Coefficient | t-Value | Coefficient | t-Value | Coefficient | t-Value | Coefficient | t-Value |
| **Age** | 0.0113 | 0.493 | 0.0108 | 0.479 | 0.0106 | 0.499 | | |
| **Edu** | −0.0140 | 0.725 | −0.0296 | 0.422 | −0.0354 | 0.352 | | |
| **Exp** | −0.0863 | 0.000 *** | −0.0677 | 0.001 *** | −0.0749 | 0.000 *** | −0.0970 | 0.000 *** |
| **Gender** | 0.3081 | 0.395 | 0.1982 | 0.561 | 0.0949 | 0.785 | | |
| **GOW** | 0.5533 | 0.038 ** | 0.5556 | 0.023 ** | 0.6080 | 0.016 ** | 0.5872 | 0.024 ** |
| **PL** | 2.0108 | 0.000 *** | 1.4167 | 0.000 *** | 1.0249 | 0.025 ** | 1.6174 | 0.003 *** |
| **Area** | 0.0370 | 0.000 *** | 0.0271 | 0.002 *** | 0.0142 | 0.237 | 0.0208 | 0.104 |
| **SWR** | 1.0450 | 0.000 *** | 1.1818 | 0.000 *** | 1.1342 | 0.000 *** | 1.0032 | 0.000 *** |
| **LYV** | 0.0003 | 0.015 ** | | | | | 0.0003 | 0.010 ** |
| **PCO** | −0.3801 | 0.140 | −0.2291 | 0.329 | −0.3013 | 0.210 | −0.5206 | 0.038 ** |
| **AI** | 0.8809 | 0.000 *** | | | | | 0.9228 | 0.000 *** |
| **PBP** | −0.3521 | 0.001 *** | −0.3267 | 0.001 *** | −0.2970 | 0.002 *** | −0.3336 | 0.001 *** |
| **HTI** | | | | | 0.0112 | 0.103 | 0.0141 | 0.062 * |
| **UPL** | | | | | 0.0444 | 0.001 *** | 0.0523 | 0.000 *** |
| **Constant** | −4.6911 | 0.001 *** | −0.7360 | 0.483 | −0.8528 | 0.429 | −4.4113 | 0.000 *** |
| **Chi-square** | 147.82 *** | | 91.29 *** | | 107.07 *** | | 165.07 *** | |
| **R-square** | 0.235 | | 0.1451 | | 0.1702 | | 0.2624 | |
| **Mean VIF** | 9.58 | | 7.02 | | 7.87 | | 7.86 | |

Notes: t statistics in parentheses. * $p < 0.05$, ** $p < 0.01$, *** $p < 0.001$.

　　The results of the model test indicate that the fitting effects of the four models are all pass the significance test. Model 1 takes the personal characteristics: age, education level (Edu), experience (Exp), gender, whether used to go out to work (GOW); production characteristic: professional level (PL), mariculture area (Area), whether sea waters registration (SWR), loss of yield value per unit area influenced by different kinds of disasters in recent three years (LYV), whether participate cooperation organization (PCO); and factors about insurance: awareness of insurance (AI) and perception of burden

level of premium (PBP) as independent variables. Compared to Model 1, Model 2 drop the independent variables of awareness of insurance (AI) and loss of yield value per unit area influenced by different kinds of disasters in recent three years (LYV). Model 3 add family characteristics: household total income (HTI) and unpaid productive loan (UPL). Considering the test results and logical consistency of the four models, which the positive and negative effects of each influencing factors, the gender, age and education level (Edu) of the fishermen are excluded from the Model 4. According to the R-squared, we find independent variables in Model 4 have higher co-explanatory power than other variables in other models.

In model 5, model 6, and model 7 variable Loss of yield value per unit area influenced by different kinds of disasters in recent three years (LYV) was break as Loss of yield value per unit area influenced by typhoon and flood (LTF), Loss of yield value per unit area influenced by drought (LDro), Loss of yield value per unit area influenced by disease (LDis), Loss of yield value per unit area influenced by pollution (LP). As we know, climate event or natural disaster will influence environmental pollution and disease spread. For example, when the climate is abnormal, shrimps are more likely to get sick, which results in more loss of yield value. In addition, climate disaster will cause a lot of pollution and drought will increase pollution accumulation. The cross-terms LDro*LDis of Loss of yield value per unit area influenced by drought (LDro) and Loss of yield value per unit area influenced by disease (LDis), and LDro*LP of Loss of yield value per unit area influenced by drought (LDro) and Loss of yield value per unit area influenced by pollution (LP) were added in model 6. The cross-terms LTF*LDis of Loss of yield value per unit area influenced by typhoon and flood (LTF) and Loss of yield value per unit area influenced by disease (LDis), and LTF*LP of Loss of yield value per unit area influenced by typhoon and flood (LTF) and Loss of yield value per unit area influenced by pollution (LP) were added in model 7. Therefore, the cross-terms are excepted to have positive impact on mariculture insurance participation.

It is consistent in Tables 5–7. As show in Table 5, climate risk, resulted by typhoon, flood and drought, have a significant positive effect on the insurance participation behavior of mariculture fishermen. However, environmental risk, resulted by disease and pollution, have no significant directly effect on the insurance participation behavior of mariculture fishermen in model 5. When we add Cross terms of climate risk (Loss of yield value per unit area influenced by drought (LDro) and Loss of yield value per unit area influenced by typhoon and flood (LTF)) and environmental risk (Loss of yield value per unit area influenced by disease (LDis) and Loss of yield value per unit area influenced by pollution (LP)) in model 6 and model 7, we can find that the impact of environmental risks becomes significant as climate risks increase (as we can see in Table 5). Due to the relatively small number of marine fishermen affected by drought, and the number of samples affected by drought and pollution or diseases is too small, the impact of cross term between drought and environmental risks on the insurance participation behavior of farmers is not significant.

**Table 5.** Logit regression results of different kinds of risks.

| Variables | Model 5 | | Model 6 | | Mode 7 | |
|---|---|---|---|---|---|---|
| | Coefficient | t-Value | Coefficient | T-Value | Coefficient | t-Value |
| LTF | 0.0004 | 0.012 ** | 0.0004 | 0.016 ** | 0.0003 | 0.099 * |
| LDis | 0.0003 | 0.266 | 0.0008 | 0.003 *** | 0.0008 | 0.001 *** |
| LDro | 0.0008 | 0.002 *** | | | | |
| LP | 0.0000 | 0.958 | | | | |
| Age | 0.0097 | 0.572 | 0.0103 | 0.548 | 0.0104 | 0.549 |
| Ede | −0.0202 | 0.631 | −0.0209 | 0.618 | −0.0230 | 0.591 |
| Exp | −0.1017 | 0.000 *** | −0.1024 | 0.000 *** | −0.1017 | 0.000 *** |
| Gender | 0.4967 | 0.209 | 0.4013 | 0.318 | 0.3679 | 0.345 |
| GOW | 0.5853 | 0.035 ** | 0.5518 | 0.049 ** | 0.5891 | 0.035 ** |
| PL | 1.4857 | 0.008 *** | 1.2397 | 0.017 ** | 1.7462 | 0.001 *** |
| Area | 0.0221 | 0.092 * | 0.0179 | 0.167 | 0.0246 | 0.062 * |
| SWR | 0.9245 | 0.001 *** | 0.9904 | 0.000 *** | 0.9523 | 0.001 *** |
| PCO | −0.3701 | 0.187 | −0.3101 | 0.272 | −0.3416 | 0.233 |
| HTI | 0.0128 | 0.096 * | 0.0112 | 0.143 | 0.0155 | 0.047 ** |
| UPL | 0.3262 | 0.002 *** | 0.3300 | 0.002 *** | 0.2922 | 0.007 *** |
| AI | 0.9343 | 0.000 *** | 0.9383 | 0.000 *** | 0.9405 | 0.000 *** |
| PBP | 0.0564 | 0.000 *** | 0.0561 | 0.000 *** | 0.0550 | 0.000 *** |
| LDro*LDis | | | 0.0000 | 0.728 | | |
| LDro*LP | | | 0.0000 | 0.317 | | |
| LTF*LDis | | | | | 0.0000 | 0.082 * |
| LTF*LP | | | | | 0.0000 | 0.055 * |
| Constant | −5.2306 | 0.000 *** | −4.6614 | 0.000 *** | −5.7143 | 0.000 *** |
| Chi square | 172.52 *** | | 172.92 *** | | 179.77 *** | |
| R-square | 0.2742 | | 0.2749 | | 0.2858 | |
| Mean VIF | 7.66 | | 7.71 | | 7.54 | |

Notes: t statistics in parentheses. * $p < 0.05$, ** $p < 0.01$, *** $p < 0.001$.

**Table 6.** Pairing comparison of willingness to participate in fishery insurance before and after financial subsidies.

| | | Participate in Fishery Insurance | | |
|---|---|---|---|---|
| | | − | + | Total |
| **Before financial subsidies** | − | 150 | 139 | 289 |
| | + | 5 | 163 | 168 |
| | | 155 | 302 | 457 |

**Table 7.** Insurance participation of Mariculture fishermen and disasters condition.

| Disasters | The Number of Fish Farmers Affected by the Disaster in Recent Three Years | Proportion (%) |
|---|---|---|
| **Typhoon and flood** | 373 | 81.62 |
| **Drought** | 155 | 33.92 |
| **Pollution** | 178 | 38.95 |
| **Disease** | 227 | 49.67 |

Source: questionnaire survey statistics in Zhejiang Province.

The McNemar test is a nonparametric statistical method for diagnosing whether there is difference in the value of the paired categorical data before and after the test. The national premium subsidy for the participation in fishery insurance is taken as an external condition, which will inevitably affect the enthusiasm of fishermen joining the insurance.

Therefore, Table 6 uses the McNemar test to analyze the willingness of fishermen's participation in the insurance without and with state subsidies. In Table 6: "−" means unwillingness and indifference; "+" means willingness. The total sample size is 457. Continuous correction was applied in this paper with a P value < 0.0001. The test found that there is significant difference between the "without subsidy" and "with subsidy" willingness, indicating that the subsidy has a clear positive incentive for the willingness of participation.

*4.2. Regression Analysis*

(1) Risk and Insurance Participation

Risk resulting in economic loss has a significant positive impact on insurance participation of mariculture fishermen. Risk is measured by the loss of yield value per unit area influenced by different kinds of disasters in recent three years that individuals are facing [60], which is an important explanatory variable for decision-making [61]. In theory, the farmers' risk is influenced by their own bounded rationality [62] and the risk atmosphere of farmers [63]. In practice, the constraints of risk can be summarized into the dual dimensions of oneself and external environment [64].

As we can see in Table 7, the number of fish farmers affected by the typhoon and flood disaster in recent three years is largest, and the disease, pollution and drought. Mariculture risk mainly includes climate risk and environmental risk. Climate risk is represented by the mariculture loss of yield value influenced by typhoon, flood and drought. Environmental risk is represented by the mariculture loss of yield value influenced by disease and pollution. Climate risk, resulted by typhoon, flood and drought, have a significant positive effect on the insurance participation behavior of mariculture fishermen. However, environmental risk, resulted by disease and pollution, have no significant directly effect on the insurance participation behavior of mariculture fishermen in model 5. The impact of environmental risks becomes significant as climate risks increase. When considering mariculture experience as technology level of marine fishermen, the longer experience of marine aquaculture, the smaller technical risks faced by fish farmers, we can make the conclusion that technological risk have a significant positive effect on the insurance participation behavior of mariculture fishermen.

(2) Awareness of Insurance and Insurance Participation

Awareness of insurance has a significant positive impact on insurance participation behavior of mariculture fishermen. Fishermen with a better understanding of the specific provisions, subsidy policies and implementation plan of fishery insurance, could understand the benefits of insurance for the systematic risk dispersion of their own production more, leading to the stronger willingness to participate in fishery insurance. Walker and Salt [65], Biggs et al. [66], and Pope et al. [67] are of the view that improving access to forecasting, early warning systems and climate information can reduce the fisheries sector's vulnerability to the changing climate. The development of early warning system is of importance for the healthy development of maricultural sector.

In addition to the financial limitations, the awareness of insurance has a significant positive influence. Risk aversion incentives are the main motivation driving fishermen to participate in the insurance. The higher systemic risk is the most important determinant of fishermen insurance participation, since mariculture is a high-input, high-output, and high-risk industry. Education and training have been identified as essential adaptation measures [68] to improve awareness of mariculture insurance of fish farmers. Education and awareness creation may help fisherfolks to make informed decisions and choices in employing appropriate adaptation measures [69]. It helps a lot make more objective and accurate decisions on the participation in fishery insurance with a correct and sufficient understanding of the insurance. As Table 8 shows, there were 56 households who did not know fishery insurance very well (including relatively unknown and totally unknown), accounting for 12.25% of the subjects, only 47 households could fully understand fishery insurance, accounting for 10.28% in 2015. There still have not any particular laws for the insurance of fishery in China, nor any policies

for regulating the mutual insurance association. A lack of legislative support and protection, the sustainable development of fishery insurance can't be guaranteed [27]. Therefore, the decision of fishermen to participate in insurance is still heavily determined by their awareness of insurance.

**Table 8.** Insurance participation of Mariculture fishermen and Awareness of insurance.

| Awareness of Insurance | 1 | 2 | 3 | 4 | 5 | Total |
|---|---|---|---|---|---|---|
| Proportion (%) | 5.03 | 7.22 | 54.92 | 22.54 | 10.28 | 100 |
| Cumulative proportion (%) | 5.03 | 12.25 | 67.18 | 89.72 | 100 | |
| Participate | 8 | 10 | 69 | 88 | 31 | 206 |
| % | 34.78 | 30.30 | 27.49 | 85.44 | 65.96 | 45.08 |
| Not participate | 15 | 23 | 182 | 15 | 16 | 251 |
| Total | 23 | 33 | 251 | 103 | 47 | 457 |

Source: questionnaire survey statistics in Zhejiang Province. Notes: Awareness of insurance: 1 = very little, 2 = relatively little, 3 = general, 4 = relatively more, 5 = very well.

(3) Perception of Burden Level and Insurance Participation

Perception of burden level has a significant negative impact on insurance participation of mariculture fishermen. The promotion of policy mutual insurance by government still needs to be improved. From the view of insurance cost, fishermen become more reluctant to participate in fishery insurance. Insurance demand is found to be negatively related to insurance premium rates [19]. At present, fishery insurance costs are mainly borne by the fishermen themselves, however, the fishermen income level is low which results in a long-term shortage of fishery insurance demand in China [70,71]. However, the insurance cost of policy-based fishery insurance, with the subsidies of governments at all levels, has not been the largest obstruction for the insurance participation of marine shrimp fishermen in Zhejiang Province.

(4) Government Subsidies and Insurance Participation

Premium subsidies are playing an important role driving fishermen to participate in insurance. The uncertainty of consumer demand, price volatility and information asymmetry pose serious market risks for marine aquaculture development [72]. Government subsidies play an important role in correcting it [29,73,74]. The effective implementation of insurance is also affected by some technical and political issues. Government subsidies for agricultural premiums have increased the enthusiasm of farmers to participate in insurance [30]. Subsidies will improve farmers' ability to pay for agricultural insurance, ease the contradiction between the low incomes of farmers and high agricultural insurance rates [11,74,75]. Studies out of China have the similar conclusion as happened in the US, the higher agricultural insurance subsidies increase the expected marginal net income of US farmers purchasing high-assured crop insurance, leading to the increase of their agricultural insurance participation rate [76]. According to the survey results, 139 households (30.42% of the total) would change their decisions to choose to participate in insurance because of the subsidies. The premium subsidy policy has an incentive effect for fishermen to participate in fishery insurance. It has increased the participation rate of fishery insurance and promoted the in-depth implementation of the fishery insurance pilot.

However, a large number of studies have proved that it is not easy for fishery mutual insurance to be perfect purely by policy support. On the one hand, the benefits brought by the increase in production with subsidies may not be able to make up for the losses suffered by the decline in market prices [76]. On the other hand, government subsidies could lead to the slack of fishermen's production and even change the motives for insurance [77].

Brunette and Couture [77] believe that the government's post-disaster relief measures have reduced farmers' willingness to pay for agricultural insurance. Although the impact of post-disaster relief on insurance demand is just on the opposite, agricultural insurance with financial subsidy

has become an important tool for farmers to enjoy economic benefits from US government [19]. The government has not played a leading role in the implementation of fishery insurance, and the local financial support system is also imperfect, with limited subsidy insurance and lack of scientific and pertinent subsidy decision [42,71].

Without adequate reinsurance or government subsidies, crop insurers would have to pass the cost of bearing the additional risk onto farmers, rendering individual crop insurance extremely, if not prohibitively, expensive [8] Agricultural insurance markets were initiated in Europe over 200 years ago in the form of privately offered protection against livestock mortality and named peril events such as crop-hail. Yet, only in the last 50 years has there been a rapid expansion and development in the range and scope of insurance products offered to producers. Most of this expansion is accounted for by an extensive range of government supports, including subsidized premiums, subsidized delivery and loss adjustment expenses, and the public provision of reinsurance services [78]. As noted, the average farmer receives approximately $1.88 in indemnities for every dollar paid in premiums [19]. By 2007, premium subsidies among high income countries totaled almost $12 billion, with the United States accounting for $3.8 billion [79].

(5)  Characteristics of Farmers and Insurance Participation

Gender, age, and education are not significant factors and basically do not have explanatory power. There are different research results on the influence of fishermen's age and education level on their participation in insurance. Some scholars have found that the age and education level of farmers have no significant influence on their willingness to participate in insurance [11]. However, some scholars also have found that the willingness of participation grows with the decrease of age and increase of education level [57]. Additionally, it is more difficult for them to accept new things, resulting in the low willingness to purchase insurance. Mariculture is based on household production mostly, in which both men and women are engaged. It is a joint decision of men and women to participate in fishery insurance. It can be seen that the gender of the head of household does not have a significant influence on the participation in fishery insurance. The main reason may be, first of all, fishermen with higher education level are mostly young and middle-aged, with strong ability to learn and master the skills. Secondly, most of the fishermen with low education level are elderly people, with relatively long mariculture time, rich production skills and experience, and strong ability of preventing climate disasters and post-disaster remediation.

(6)  Mariculture Experience and Insurance Participation

Mariculture experience has a negative influence on fishermen's insurance participation. The imperative of climate event requires increased capacity of farmers to make both short- and long-term planning decisions and technology choices [80]. The longer fishermen are engaged in mariculture, the more experience and methods for culture they have. Experienced fishermen are reluctant to change the existing ways to avoid risks, and do not believe in the protection of other organizations for their own production. Employing the use of traditional ecological knowledge management systems as an adaptation measure proves to be very useful as it imparts additional knowledge and perspectives based on locally developed practices such as fish species management.

(7)  Labors Used to Go out to Work and Insurance Participation

As Table 9 shows, the proposition of insurance participation of labors used to go out to work is lower than that of labors not used to go out to work. Advanced technology is not conducive to the development and promotion of fishery insurance to a certain extent. Some scholars have found that farmers could prevent and share risk through differentiated planting, adopting conservative technology, reciprocal credit, going out to work, or doing business [13]. Labors used go out to work have a significant positive influence on insurance participation. Labors used go out to work are often younger and educated, which could be a help for them to understand the risks brought by production

and the important role of fishery insurance. Meanwhile, migrated workers with cash earning will relieve the burden of mutual insurance payment.

**Table 9.** Mariculture fishermen used to go out to work and insurance participation.

| Whether Used to Go Out to Work | 0 | 1 | Total |
|---|---|---|---|
| Participate | 181 | 70 | 251 |
| Proportion (%) | 56.56 | 51.09 | 54.92 |
| Not participate | 139 | 67 | 206 |
| Total | 320 | 137 | 457 |

Source: questionnaire survey statistics in Zhejiang province. Notes: 0 = not used to go out to work, 1 = used to go out to work.

(8)　Production Specialization and Insurance Participation

Total household income has a positive influence on fishery insurance participation. Holthausen and Baur [58] found that household income structure has a significant influence on farmers' participation behavior. The production specialization, measured by the proportion of income from mariculture to the total income has a positive influence on the participation of aquaculture insurance. The higher the proportion, the greater the importance of mariculture to fishermen's families. It is more necessary and urgent for specialized fishermen to spread production risks than part-time fishermen. Therefore, fishermen with higher degree of specialization have more enthusiasm to participate in fishery insurance. When agricultural income is not the main source of income, farmers' willingness to purchase agricultural insurance is not strong. Not only the income structure, but also the total income level will affect the willingness of farmers to purchase insurance [57].

(9)　Farming Scale and Insurance Participation

As Table 10 shows the proportion of marine aquaculture area greater than 30 mu is higher than that of marine aquaculture area less than 30 mu. The commercial companies prefer the large-scale fishery and aquaculture companies to those small-scale ones which, therefore, need the support from government, especially when they suffer great economic losses [27]. The mariculture area has a significant positive influence on fishermen's participation behavior. Goodwin [81], Goodwin and Smith [82] found that agricultural insurance is inelastic relative to the area insured [81,82]. consisting with the finding of Ning, Li, and Zhong, the total cultivated land area is an important factor affecting farmers' purchase of agricultural insurance [30]. Fishermen with larger aquaculture areas are bearing larger system risks, leading to higher willingness to participate in insurance. However, in the regression of Model 3 and Model 4, when the total household income and the household unpaid production loan were added to the independent variables, the influence of the size of mariculture area on the fishermen's willingness to participate in insurance became insignificant. The possible reason is, fishermen are always paying attention to the guarantee of overall income and risk reduction.

**Table 10.** Insurance participation of Mariculture fishermen and mariculture area.

| Area (mu) | 0~10 | 10~20 | 20~30 | 30~40 | 40~50 | 50~60 | 60~70 | 70~80 | 80~90 |
|---|---|---|---|---|---|---|---|---|---|
| Participate | 3 | 58 | 51 | 72 | 4 | 4 | 6 | 4 | 4 |
| Proportion (%) | 18.75 | 34.94 | 38.93 | 67.29 | 57.14 | 57.14 | 66.67 | 50.00 | 66.67 |
| Not participate | 13 | 108 | 80 | 35 | 3 | 3 | 3 | 4 | 2 |
| Total | 16 | 166 | 131 | 107 | 7 | 7 | 9 | 8 | 6 |

Source: questionnaire survey statistics in Zhejiang Province.

(10)　Aquaculture Registration and Insurance Participation

The implementation of the policy of aquaculture registration has promoted the participation of insurance (see Table 11). Compared with the lease of short-term contracts, fishermen who have

registered the mariculture areas are more concerned about the long-term development of production. In order to ensure stable and sustainable development, fishermen are more willing to participate in mutual insurance. Essentially, the contract is a commitment on behaviors of the parties on both sides, and also an arrangement of bilateral coordination regarding behaviors [83]. As the bond of market transactions, the contract is the institutional arrangement regarding the rights and obligations of the assets. Incomplete contract arrangements of farmland property rights are another issue that has attracted lots of attention in China [84]. Under the household contract responsibility system, farmers were given the residual rights of land use and usufruct [84]. However, the government intervened heavily on rights allocation by making frequent land adjustments [84], which has led to incomplete farmland property rights [85] and then damaged farmer's long-term intentions of investment in land [84]. In the evolution of China's farmland system, although the residual control rights and the residual claim rights of farmlands have been gradually relaxed by the government [86], farmland property rights should be further clarified [87].

**Table 11.** Insurance participation of mariculture fishermen and registration of sea waters.

| Whether Sea Waters Registration | 0 | 1 | Total |
|:---:|:---:|:---:|:---:|
| Participate | 125 | 81 | 206 |
| Proportion (%) | 37.65 | 64.80 | 45.08 |
| Not participate | 207 | 44 | 251 |
| Total | 332 | 125 | 457 |

Source: questionnaire survey statistics in Zhejiang Province. Notes: 0 = not registration; 1 = Registration.

(11)  Income and Insurance Participation

The impact of total income on fishermen's willingness to participate in insurance is not obvious when considering the other fishermen families characteristics. When these factors are neglected, the total income becomes a significant positive factor for the fishermen's willingness to participate in insurance. One possible reason is that the pressure to pay for insurance will be less stressful for fishermen with higher total income. In order to ensure the stability of mariculture production and the benefits of farming, fishermen with higher incomes would be more willing to participate in fishery insurance.

(12)  Unpaid Productive Loan and Insurance Participation

Unpaid productive loan has a significant positive impact on mariculture insurance participation. High risks in fishery production, as well as other issues related to payment, make it hard for some financial institutions to offer loans to fishery. In order to minimize the risks, the financial institutions prefer to provide loans to those with insured assets. That means, without insurance, the financial institutes would be hesitate to offer the loans, which limits the fishermen input and the development of fishery [27].

Unpaid productive loans will also promote fishermen's participation in fishery insurance (see Table 12). There is more risk on production based on loan than that based on fishermen's own property under the same condition. The fishermen also have a lower ability of diversifying risk and keeping sustainable development. Fishermen with more unpaid productive loans have a stronger need to participate in fishery insurance under the same production conditions. Therefore, it is necessary to establish and enhance the direct relations between the fishery insurance and the financial credit, because the fishery insurance could not only effectively increase the investment from some financial institutions, but also encourage the fishermen's input and their adoption of new technique to improve the productivity and their payment capacity [27].

**Table 12.** Insurance participation of mariculture fishermen and unpaid productive loan.

| Unpaid Productive Loan | 0~10 | 10~20 | 20~30 | 30~40 | 40~50 | 50~60 | 60~70 |
|---|---|---|---|---|---|---|---|
| Participate | 149 | 36 | 7 | 10 | 2 | 2 | 0 |
| Proportion (%) | 40.71 | 60.00 | 46.67 | 90.91 | 100.00 | 100.00 | 0.00 |
| Not participate | 217 | 24 | 8 | 1 | 0 | 0 | 1 |
| Total | 366 | 60 | 15 | 11 | 2 | 2 | 1 |

Source: questionnaire survey statistics in Zhejiang Province.

(13) Participating in Cooperative Organization and Insurance Participation

Participating in cooperative organization has negative impact on mariculture insurance participation. As Table 13 shows, the most involved organizations of marine farmers are cooperatives, except for mutual insurance association. The participation rate of marine farmers who join cooperatives is only 33.06%. The fishery in developing countries is mostly managed by fishermen or small-scale fishing farmers [88], who will suffer great losses by climate disasters and who should be protected by the fishery insurance. But due to the fact that most of these fishermen or farmers often live and work separately, they find it hard to get their preferred insurance from the private insurance companies. Then the fishery cooperatives and mutual associations would come to their aids. The cooperatives and associations can organize those fishermen together and offer them a risk diversification project to reduce their risks of losses brought by climate disasters [27]. The main functions of the National Federation of Fishermen's Cooperative are to arrange the input of fishery and the supply of facilities, to transfer the technical skills to fishermen, to provide insurance to fishermen, to raise and sell fish, to provide consulting services, to provide training, to promote fishery export, and communicate with the government and related departments [27].

**Table 13.** Insurance participation of Mariculture fishermen and membership of organizations.

| Participate in Cooperative Organization | 1 | 2 | 3 | 4 |
|---|---|---|---|---|
| Participate | 41 | 173 | 0 | 31 |
| Proportion (%) | 33.06 | 100.00 | 0.00 | 100.00 |
| Not participate | 83 | 0 | 19 | 0 |
| Total | 124 | 173 | 19 | 31 |

Source: questionnaire survey statistics in Zhejiang Province. Notes: 1 = Cooperatives; 2 = Mutual insurance association; 3 = Cooperate with the company; 4 = Alliance between large farmer households.

Mariculture technology risks are mainly manifested in two aspects: technical shortage and application deviation [72]. Due to the slow progress or even the lack of formal agricultural finance and insurance in China, the contract of production, processing and sale with firms are the commonly adopted approaches for farmers to avoid agricultural risks [89]. Participating organizations can help fishermen to improve production technology and management level, provide excellent breeding varieties, and purchase high-quality and low-cost production materials, thereby improving product quality, reducing market information asymmetry, and ensuring product sales. We can see that participating in aquaculture cooperative economic organizations can reduce the systemic risks faced by fishermen a lot and reduce the enthusiasm of fishermen for insurance.

**5. Conclusions**

Risk is one of the most important determinants of promoting the participation of marine fish farmers in insurance. Technical risks and climate risks directly and effectively promote insurance participation of marine fishermen. Environmental risks have no directly significant impact on insurance participation of marine fishermen. But as the climate risk increases, the impact of environmental risks on the insurance participation behavior of farmers becomes significant.

Fishermen's awareness of insurance in the past years has influenced individual decisions on insurance participation. It is important to enhance fishermen's awareness and effectiveness of insurance to promote their insurance participation. Firstly, the relevant departments of the grassroots government should widely publicize the significance, insurance methods, and policy measures of fishery insurance to enhance the insurance perception of fishermen and create a good social atmosphere. Secondly, they should popularize insurance knowledge, and explain insurance contracts, compensation standards, and scope of responsibility in various easy-to-understand forms, especially the typical claims of fishery insurance that occurred in the past, from which fishermen could really believe that great benefits are available with small amount of money spent so that insurance is deeply rooted in the hearts of the people as a scientific and effective way to avoid risks.

The Burden of Insurance premium is an important block for mariculture fishermen to participate fishery insurance. Increasing income of mariculture fishermen and decreasing insurance premium are useful ways to stimulate mariculture insurance participation. The characteristics of public goods of fishery mutual insurance determine that the development of fishery insurance cannot be separated from government support. Government subsidy is necessary to promote insurance participation, especially for the fishermen with low level income.

Stimulating large-scale specialized aquaculture production development, which can not only promote fishermen to participate in fishery insurance, but also improve the overall production efficiency and upgrading of industrial structure of aquaculture production in Zhejiang Province. While fundamentally ensuring the stability and sustainable development of mariculture, to promote large-scale mariculture and specialized production will reduce the cost of aquaculture production and systemic risks, and enhance the competitiveness of the marine products of in the international and domestic markets.

Promoting the sustainable development of mariculture production, especially the stability of water use. At present, the sea area registration plan has been launched, but the implementation of this policy is not in place and not popular enough. Most of the aquaculture waters have not been registered, many fishermen are not clear about the specific content of the registration and whether the sea areas they have cultivated have been registered. This situation limits the stability of mariculture, and also reduces the enthusiasm of fishermen to participate in insurance and diversify system risks. The government needs to further promote the sea area registration system, clarify the specific sea location and area that could be used by fishermen, and clarify the rights and responsibilities of the fishermen in their aquaculture waters.

In addition, giving more preferential treatment to special financial loans to mariculture farmers and encourage them to expand the scale of loans, which can not only stimulate mariculture production development, but also promote farmers to enhance risk awareness and insurance participation willingness.

**Author Contributions:** It should be noted that the whole work was accomplished by the authors collaboratively. All authors read and approved the final manuscript.

**Funding:** This research was funded by the Major Program of the National Social Science Foundation of China, Grant No.14 ZDA070 and the Fundamental Research Funds for the Central Universities, Grant No. 2018.

**Conflicts of Interest:** The authors declare no conflict of interest.

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
