# Peer review of "Systemic Risks of Climate Events and Households’ Participation in Mariculture Mutual Insurance: A Case Study of Shrimp Producers in Zhejiang Province"

_sustainability, doi:10.3390/su11041164_

Reviewer 1 Report

The authors should clarify the rationale behind the choice of variables cited in 238-240. There is no reference and the list does not have a strong scientific rigor. Since this is the core of the following calculations and approaches, it needs to be clarified. Please expand and motivate.

There is no sample size calculation with respect to population, and thus it is not clear if the results have any general value, or they can only be considered as a statistical exercise.

When a questionnaire is considered effective? And when it is not? (line 271-onwards).

The description of Table 2 is pretty imprecise with respect to what is available in it.

I found really hard to follow Table 4 and Table 5. The name of variables should be clarified at the beginning in an acronym section. The differences between the Models (1,2,3,4) should be clearly stated and summarized before Table 4.

The authors claim the Model 4 performs better than the other models, and it seems due to the large number of statistically meaningful p-values. Is this the case? If yes, why? If not, please clarify why Model 4 is better than the other and provide a critical reflection to explain why it is really better.

Same for Table 5, I cannot really follow how does it relate to Table 4, and to the rest of the mathematical formulation.

In Table 7, it is not specified if the values refer to the entire country, or just the region under investigation.

Table 8 is not linked in the text. How is the “knowledge of insurance” defined in five levels? Which categories and rules define the structure of the variable?

Generally speaking there is limited reflection on the obtained results and on the real nature of risks.

The authors claim the work is about the “perception of systemic risks”, but they miss to give a definition of what is considered to be a “risk” in this research, and how the risk dimensions are significant to represent its nature..

Since the paper is highly dependent on the geography of the country, I would recommend adding a geographical map, specifying the focus of the analysis and the sub-region (as for he info in 271-272)

Author Response

A point-by-point response is in the word file. Please check it. Thanks.

Reviewer 2 Report

The authors present the results from a survey about mutual insurance for shrimp farmers in the Zhejiang province. A regression analysis shows the likelihood to participate as a function of characteristics.

General comments:

-) The study and the survey are an interesting depiction of the fish farming industry of the province.

-) The title is a bit misleading in the sense that climate change is not tackled by the survey. It seems perception about climate change  by farmers was not covered in the survey. Sounds more like a catchy title rather than what is actually studied. Maybe the title should be changed to reflect the actual content of the study: participation in mutual insurance.

-) The policy implications of the conclusion cannot be drawn from the existing study. Existing government intervention has been mostly quantified by a McNemar test. The test only highlights than people are ready to subscribe more to insurance when their cost drop, which is not a surprise. The regression profiles the likelihood to subscribe to an insurance with only a 26% R2. Furthermore, there is no element in the study to justify direct government intervention. To the contrary, the study has highlighted the precise elements that make people more likely to participate on their own. No element related to externalities has been unveiled by the study that could possibly support the need for government to support shrimp culture in the form of mutual insurance rather than anything else aimed at promoting the industry such as a disaster relief program or any other policy that could mitigate disasters. The authors point that the degree of education is not a factor, revealing that information about the risks may not be the problem. Authors mention otherwise in the conclusion (607-610) but no data back these statements. All conclusions related to competitiveness or infrastructure are opinions not directly supported by the study or the survey. It is important that the conclusion be fully supported by the results of the study and not loosely related. It may very well be that participants are not interested that much into mutual insurance, which is a perfectly valid result to the study.

Specific comments:

45: Cost of what is lower than insurance? Damage?

64: It is not clear that [29] demonstrates that participation in mutual insurance is “key”, please rephrase

98: please cite source of numbers used to describe production growth

106 please rephrase the sentence starting with “And”

113: Figure 1, please cite sources (China fishery statistical yearbook —> which year, author, etc.). Also on all other figures and elsewhere in the text

131-139 + Figure 4: I understand the authors want to highlight the importance of Zhejiang in terms of fisheries, however, since the context of the paper is about mariculture, please display the proportion of mariculture output rather than fisheries (per region)

186-196: Do we have any data on the amount of claims per damage category?

210: where does the sentence end (check punctuation)

216: cite sources for the 40%

222: please include an introduction to the whole methodology: what it is you want to measure, how you did it: first, a survey to get data, then analyzing the data with a regression.

224: Please use same standard citation method across the document

231: “Equation (1)” (not Formula 1)

247: Please explain how Knowledge of insurance is measured

262: I guess it is too late, but questions about perception on climate change should have been included in the questionnaire. Please justify why you can draw conclusions on perception on climate change without controlling for it in the survey

273: Please provide the exact effective rate (89.6%)

290: Which source is used to state that the sample is representative of the marine farmers population?

291: The abbreviation system is a bit confusing, there is a lot of back and forth in reading the table 4 with table 2 in order to understand what it is about. Can you just use more explicit variable names in table 4? Also, I could not find the definition of variables TP and POL in table 5

315: Typo in the column name “Mode 3”. Variable LOS is significant with a 0.000 parameter. Can you precise the actual value in scientific notation? 

318: Not sure what “ideal” means in this context. Can you detail?

322: Table 4 and Table 5 should actually be the same table? Why is the formatting different? 

Please review the whole presentation of these results. 

322: Model-wise, I am very unsure of the robustness of the model. Can you please add some diagnostics (try to assess quasi-collinearity in particular, eg, I do not see how age and experience are not strongly correlated). 

322: What do you make of the fact that the model has only 26% R2, and also, I could not find the R2 for models of table 5

334: I do not understand the conclusions at all. What are the cross-terms? How to interpret them? Why even crossing these terms? Why do you reach the conclusion that environmental risks becomes significant as climate risks increase? This is a strong statement that makes no sense to me in the context of the regression without clear explanations. It is a very important element of the study that requires more detailed attention.

365: please keep citation standards

429: Please cite the studies you mention

549: the notion that shrimp farmers live away from each other has not been covered in the survey or the analysis

584-615: Please change to conclusion to match actual results from the study, or include quantified information that backs the conclusion.

Author Response

(The authors gave the same response as above.)

Reviewer 3 Report

The research is focused on the factors determining fish farmers’ participation in mariculture insurance. The Authors performed a regression analysis between independent variables and the fishermen participation in fishery insurance to investigate their relationships.

The Introduction section provides a sufficient background and the research design is appropriated. Conversely, Methods and Results must be improved.

Overall, the research is original and really interesting and could be of interest to the readers. For this reason, the manuscript can be considered suitable for the publication after few major revisions.

In particular, it is not clear how the Authors selected the variables for the regression model (not referring to the bibliographic selection), or if they preformed a selection. If so, how did they select the variables? Have the Authors evaluated if all the variables could be used by the regression model? Maybe performing an univariate comparison between the dependent variable and every single independent one could help to evaluate which variables could be usede in the multivariate analysis.

Tables no.4 and no.5 are not clearly presented. The Authors should improve them, specifying the p-value, the OR and IC in the headings. This correction and modification  is pivotal for the comprehension of the results. Moreover, the abbreviation of the variables names could be omitted, keeping the not abbreviated form. 

I strongly suggest to better clarify the statistical analysis performed (software, procedure, etc...), revising at the same time the quality of results presentation.

Author Response

A point-by-point response is in the word file. Please check it. Thanks.

Round  2

Reviewer 1 Report

The authors addressed successfully the comments I raised.

Author Response

Thank you very much for your review.

Reviewer 2 Report

Thank you for adjusting the document according to suggestions.

Maybe reconsider the title given that climate change is only loosely related to the study that is really about insurance for any type of climate event.

Author Response

 A point-by-point response to the your comments is in the word file. Please check it. Thanks.

Reviewer 3 Report

All the issues raised have been addressed by the Authors

Author Response

Thank you very much for your review.